# Reshaping the Gut: Symptoms, Nutrition and Microbiota After Bariatric and Endoscopic Procedures in Obesity

**DOI:** 10.3390/nu18010108

**Published:** 2025-12-28

**Authors:** Tommaso Mancuso, Claudia Di Rosa, Alessia Falcone, Laura Restaneo, Nicolò Citterio, Dario Biasutto, Simone Carotti, Mentore Ribolsi, Annamaria Altomare, Michele Cicala, Michele Pier Luca Guarino

**Affiliations:** 1Research Unit of Gastroenterology, Department of Medicine and Surgery, Università Campus Bio-Medico di Roma, Via Alvaro del Portillo, 21, 00128 Roma, Italy; tommaso.mancuso@unicampus.it (T.M.); laura.restaneo3@gmail.com (L.R.); m.ribolsi@unicampus.it (M.R.); m.cicala@unicampus.it (M.C.); m.guarino@unicampus.it (M.P.L.G.); 2Operative Research Unit of Gastroenterology, Fondazione Policlinico Universitario Campus Bio-Medico, Via Alvaro del Portillo, 200, 00128 Roma, Italy; 3Research Unit of Food Science and Human Nutrition, Department of Sciences and Technologies for Sustainable Development and One Health, Università Campus Bio-Medico di Roma, Via Alvaro del Portillo, 21, 00128 Roma, Italy; c.dirosa@unicampus.it; 4Department of Sciences and Technologies for Sustainable Development and One Health, Università Campus Bio-Medico di Roma, Via Alvaro del Portillo, 21, 00128 Roma, Italy; falcone.alessia@outlook.it; 5Fondazione Policlinico Campus Bio-Medico, Therapeutic GI Endoscopy Unit, Via Alvaro del Portillo 200, 00128 Roma, Italy; citterion@gmail.com (N.C.); d.biasutto@policlinicocampus.it (D.B.); 6Microscopic and Ultrastructural Anatomy Research Unit, Department of Medicine and Surgery, Università Campus Bio-Medico di Roma, 21, 00128 Roma, Italy; s.carotti@unicampus.it

**Keywords:** obesity, bariatric surgery, nutrition, bariatric endoscopy, microbiota

## Abstract

Obesity is a multifactorial disease linked to chronic inflammation, metabolic disorders, and gut microbiota dysbiosis. Bariatric surgery (BS) and endoscopic sleeve gastroplasty (ESG) are effective for sustained weight loss and comorbidity improvement but may cause gastrointestinal and nutritional complications. This narrative review, informed by a structured literature search, synthesizes evidence on gastrointestinal side effects, gut microbiota alterations, and nutritional management after BS and ESG. Literature searches in PubMed and Scopus, without time limits, included English full-text articles on postoperative symptoms, microbiota changes, and nutritional outcomes. Bariatric procedures (e.g., Roux-en-Y gastric bypass, sleeve gastrectomy) and ESG are associated with adverse events such as dumping syndrome, GERD, nausea, and micronutrient deficiencies. Surgery induces profound shifts in gut microbiota composition and diversity, contributing to improved metabolic regulation. ESG, though less invasive, produces comparable microbial changes with a favorable safety profile. Nutritional management—progressive diet protocols and supplementation—is critical for preventing deficiencies and sustaining outcomes. Mediterranean-style diets appear more sustainable than high-protein regimens. Study heterogeneity, small cohorts, and limited long-term ESG follow-up reduce generalizability. Multidisciplinary care integrating surgical or endoscopic approaches with personalized nutrition and microbiota modulation is essential to optimize outcomes in obesity management.

## 1. Introduction

Obesity is a multifactorial condition characterized by excessive accumulation of body fat and is often associated with a range of metabolic disorders, including type 2 diabetes (T2D), cardiovascular diseases, and metabolic dysfunction-associated fatty liver disease (MAFLD). It arises from a combination of genetic, environmental, and lifestyle factors, and is often associated with significant alterations in gut microbiota. In individuals with obesity, gut microbiota dysbiosis is commonly observed and is characterized by reduced microbial diversity, alterations in the Firmicutes/Bacteroidetes ratio, and changes in taxa associated with low-grade inflammation and impaired metabolic homeostasis. These alterations contribute to energy imbalance, systemic inflammation, and metabolic dysregulation [1,2,3,4].

The primary goal of obesity management and treatment is improving health status through weight loss of more than 10% of total body weight. Like all chronic diseases, the management of obesity requires a long-term, multidisciplinary approach that takes into account the therapeutic goals of each individual and the benefits and risks of various therapies. The combination of lifestyle interventions (diet and physical activity), anti-obesity drugs, and bariatric surgery (BS) promotes lasting weight loss and associated health benefits [5].

Although lifestyle interventions and drug therapies can help achieve weight loss, their effects are often counteracted by compensatory physiological responses, predisposing individuals to weight relapse. The most successful long-term strategy remains bariatric and metabolic surgery (BMS), such as Sleeve Gastrectomy (SG) and Roux-en-Y gastric bypass (RYGB), which enable patients to lose between 50% and 75% of excess body weight within one year [6].

The 1991 National Institutes of Health (NIH) Consensus Statement recommends that patients being considered for bariatric surgery be evaluated by a multidisciplinary team with access to medical, surgical, psychiatric, and nutritional expertise [7]. According to the 2022 ASMBS/IFSO guidelines [8], eligibility criteria for bariatric surgery differ depending on whether obesity is accompanied by metabolic disease. Bariatric surgery is recommended for individuals with a BMI ≥ 35 kg/m^2^, regardless of the presence or severity of comorbidities. In individuals with metabolic diseases, such as type 2 diabetes, surgery should also be considered for those with a BMI between 30 and 34.9 kg/m^2^, particularly when metabolic control is inadequate despite optimal medical therapy [8].

Bariatric surgery procedures like SG and RYGB, induce significant anatomical and physiological changes, leading to both therapeutic benefits (such as weight loss and metabolic improvements) and adverse effects, particularly on gastrointestinal function [1]. Bariatric surgery alters the GI tract’s structure, reducing nutrient absorption and impacting hormonal pathways [1].

In recent years, Endoscopic Sleeve Gastroplasty (ESG) has emerged as a minimally invasive, incisionless, and reversible alternative to bariatric surgery. ESG creates a restrictive gastric sleeve through endoscopic full-thickness suturing, leading to significant weight loss with a low adverse event rate and preserving gastric anatomy [1,4,9]. Both bariatric surgery and ESG influence gut microbiota composition and metabolic outcomes, albeit through different mechanisms. Recent studies have shown that gut microorganisms contribute to the regulation of energy homeostasis and fat mass accumulation and may play a role in the onset of obesity and the development of its complications [10,11,12,13]. The characterization of the gut microbiota in individuals with obesity is still not well defined. However, it is known that obesity is associated with gut microbiota dysbiosis, characterized by a reduction in alpha diversity and an increase in the *Firmicutes/Bacteroidetes* ratio [10]. Emerging evidence highlights the role of gut microbiota in the pathophysiology of obesity, suggesting that microbiota modulation may be a promising therapeutic target. Strategies such as dietary interventions, probiotics, and prebiotics aim to restore microbial balance, but surgical or endoscopic interventions can also have an important impact [3,9]. Numerous studies have demonstrated that gut microorganisms influence host energy homeostasis, lipid accumulation, glucose metabolism, and inflammatory signaling, thereby contributing to the development and persistence of obesity. Specific microbial communities promote energy extraction from indigestible polysaccharides, regulate short-chain fatty acid (SCFA) production, and modulate enteroendocrine hormones such as GLP-1, PYY, and ghrelin [10,11,12,13]. Dysbiosis has also been associated with metabolic endotoxemia and chronic low-grade inflammation [14,15], whereas restoration of microbial diversity and SCFA signaling improves metabolic regulation and reduces adiposity [16,17,18].

The aim of the present review is to examine gastrointestinal (GI) symptoms arising from bariatric surgery and ESG and the most appropriate nutritional approaches before, during, and after the interventions.

## 2. Methods

The PubMed and Scopus databases were consulted using the following search terms: “bariatric surgery”, “endoscopic sleeve gastroplasty”, “obesity”, “gut microbiota” alone or in combination with “diet”, “nutritional approach”, “Mediterranean diet”, “side effects”, and “predictive factors”. The search was focused on full-text papers published in English with no date restrictions.

Studies were included if they addressed gastrointestinal symptoms, nutritional aspects, or gut microbiota alterations after bariatric or endoscopic procedures. Exclusion criteria were conference abstracts, case reports, and articles not related to postoperative outcomes. Approximately 430 records were identified, and after screening titles and abstracts, 138 full-text articles were assessed for eligibility; a total of 102 studies were ultimately included in this narrative synthesis.

## 3. Bariatric and Endoscopic Procedures

### 3.1. Bariatric Procedures

Bariatric surgery is classified in:“Malabsorptive procedures” (biliopancreatic diversion (BPD) with or without duodenal switch (BPD-DS), or a combination of both (Roux-en-Y gastric bypass (RYGB): they bypass part of the small intestine“Restrictive procedures”: laparoscopic adjustable gastric banding (LAGB) and laparoscopic sleeve gastrectomy (LSG): they are based on decreasing the size of the stomach

Historically, weight loss was thought to be responsible for improvement in metabolic comorbidities; however, there is now ample evidence that improvement in the metabolic profile is observed as early as days after surgery, before significant weight loss. Similarly, it is understood that the mechanisms of action of these procedures are more sophisticated than simple structural anatomical changes, as there appears to be a rapid alteration in the pattern of intestinal hormones that affect appetite, satiety, intestinal motility, and glucose and lipid homeostasis. In addition, changes in bile acid concentration and gut microbiota are believed to play an important role [19].

The BPD, as developed by Scopinaro et al. [20], is a staged surgical technique involving distal gastrectomy with removal of approximately two-thirds of the stomach, including the pylorus, along with cholecystectomy and the actual biliopancreatic diversion. This diversion creates a dual intestinal tract to delay the mixing of the ingested food with biliopancreatic secretions. Importantly, no segment of the small intestine is removed. Instead, two separate anastomoses are constructed—one for digestive flow and another one for biliopancreatic secretions—resulting in separation of the food bolus from digestive enzymes until they meet approximately 50 cm proximal to the colon [20]. The Duodenal Switch variation, also known as biliopancreatic diversion with duodenal switch, involves vertical rather than horizontal gastric resection, thus preserving the pylorus [19]. RYGB entails the formation of a small gastric pouch (approximately 20–50 mL in volume), which is proximally anastomosed to a jejunal limb. This limb is then distally connected to the duodenum, thus bypassing the gastro-duodenal segment and its associated digestive and absorptive processes [19].

Among restrictive techniques, LSG has been the most performed bariatric surgical procedure worldwide since 2014 [21], and it was initially introduced and validated as a bariatric surgical treatment for obesity by Gumbs et al. in 2007 [22]. LSG entails the resection of approximately 75% of the stomach, including the entire fundus, greater curvature, and most of the gastric body, resulting in a permanent tubular formation of the gastric lumen [23]. In terms of invasiveness, LSG falls between LAGB and malabsorptive procedures. It can be performed either as a definitive standalone intervention for patients with class I or II obesity or as an initial step within a step-up therapeutic approach, particularly in cases of severe obesity (e.g., BMI > 60 kg/m^2^). The procedure can be subsequently revised or converted into a gastric bypass or BPD-DS to broaden therapeutic efficacy [23].

Compared to malabsorptive surgeries, LSG is less invasive as it: (1) maintains physiological absorption of macro- and micronutrients, (2) avoids the need for anastomoses, thereby reducing intra- and postoperative risks and (3) minimizes the risk of dumping syndrome through preservation of the antrum and pylorus [24].

A procedure that combines both restrictive and malabsorptive components is the Mini Gastric Bypass, or One-Anastomosis Gastric Bypass. It involves the creation of a long, narrow gastric pouch (150–200 mL in volume, at least 15 cm in length), which is anastomosed to a segment of intestine approximately 150–200 cm distal to the ligament of Treitz. This allows the food bolus to bypass the gastric antrum, duodenum, and proximal small bowel, thereby inducing malabsorption [25].

### 3.2. Endoscopic Procedures

The most known endoscopic procedures are the intragastric balloon (IGB) and the ESG. The IGB is a temporary and minimally invasive therapy for weight loss and is currently considered the primary option for patients with mild obesity. As a space-occupying device, it reduces gastric capacity, resulting in decreased hunger and food intake. Several types of balloons are available, filled either with liquid or air. The most used is the non-adjustable liquid-filled balloon, due to its lower complication rate. The mechanism of action is multifactorial and includes both physiological and neurohormonal changes. The device acts as an artificial bezoar, occupying space in the stomach and promoting early satiety [26]. Air- and fluid-filled balloons remain in place for 3 and 6 months, respectively [27,28,29]. The therapeutic efficacy of IGB treatment is considered moderate, with a non-negligible rate of complications.

ESG, also known as gastric volume endoscopy (GVE) or Vertical Endoscopic Gastroplasty (VSG), is a minimally invasive, reproducible, and reversible procedure. Since its initial description, ESG has emerged as the primary endoscopic treatment for patients who are either not suitable candidates for bariatric surgery or who decline surgical intervention, effectively replacing the IGB as the standard endoscopic approach for obesity management [1,30]. The procedure involves tubularization of the gastric lumen by placing a series of interrupted or continuous sutures, leading to an estimated 70% reduction in gastric volume [31]. Sutures are placed along the greater curvature of the stomach, extending from the anterior wall to the posterior wall, starting at the incisura angularis or the body-antrum junction and continuing up to the fundus-body junction [32]. ESG is created using a commercially available endoscopic suturing device which requires the use of a dual-channel therapeutic gastroscope to operate. Full thickness suture placement is facilitated by the use of a tissue helix device which captures the targeted suture placement site on the gastric wall and retracts it into the suturing arm of the device [1,33]. Compared to LSG, ESG preserves the native gastric anatomy, as it does not involve tissue resection and does not affect the gastric fundus.

## 4. Predictive Factors of Weight Loss Outcomes After Bariatric Surgery

Weight loss outcomes after bariatric surgery are determined by a multifactorial interaction between baseline anthropometric characteristics, demographic variables, metabolic status, psychological profile, surgical technique, and postoperative management. In particular, preoperative BMI [34,35], age [36,37], and sex [38] may modulate the magnitude of weight loss, while the presence of type 2 diabetes and psychological disorders can negatively affect long-term results [39,40]. Furthermore, the type of bariatric procedure and adherence to structured, multidisciplinary follow-up programs play a pivotal role in sustaining weight loss over time [41,42,43]. These key predictive factors and their relative influence on postoperative outcomes are summarized in Figure 1.

### 4.1. Baseline Body Mass Index (BMI)

Patients with a very high initial BMI tend to lose more absolute weight (in kilograms), yet they often maintain a higher postoperative BMI compared to those starting from a lower weight. Baseline BMI has consistently emerged as the most influential single predictor of postoperative nadir weight. In fact it is associated with lower percentage of Excess BMI Loss (EBMIL) in both sexes [34]. This highlights the importance of contextualizing weight loss expectations based on preoperative BMI values [35].

### 4.2. Age and Gender

Although the relationship between age and bariatric surgery outcomes remains debated, several studies suggest that increased age is negatively associated with weight loss. Younger patients tend to achieve significantly better results, while older individuals often experience smaller reductions in weight and a higher risk of long-term weight regain [36]. A large nationwide registry study by Dreber et al., involving over 2000 patients with 5-year follow-up, confirmed significantly greater weight loss among younger patients [37]. Sex does not have a clear effect on efficacy outcomes of bariatric procedures [38]. Variability across studies may be attributed to sample size imbalances and methodological differences.

### 4.3. Presence of Type 2 Diabetes

Type 2 diabetes is frequently associated with elevated insulin levels, which can hinder weight loss. Hyperinsulinemia promotes lipogenesis and suppresses lipolysis, contributing to fat accumulation. Moreover, insulin’s anabolic properties, caloric intake adjustments to prevent hypoglycemia, and reduced glycosuria after surgery may all explain why patients with diabetes often lose less weight compared to subjects not affected by type 2 diabetes [39].

### 4.4. Psychological Disorders

Psychological conditions such as depression and binge eating disorder (BED) are prevalent among bariatric surgery candidates. While their short-term impact on weight loss may be limited, long-term outcomes are often compromised. Emotional instability and disordered eating patterns after surgery can hinder sustained weight loss. Early diagnosis and structured postoperative psychological support are essential [40].

### 4.5. Type of Procedure

The choice of bariatric procedure plays a crucial role in long-term weight maintenance. ESG, SG and RYGB yield comparable short-term outcomes. However, over time, RYGB has demonstrated greater durability [41]. These differences highlight the importance of evaluating long-term efficacy when selecting the surgical approach.

### 4.6. Adherence to Follow-Up

Participation in a structured, multidisciplinary follow-up program is a well-recognized predictor of improved weight loss outcomes—especially in patients undergoing ESG [42]. Patients who adhere to follow-up visits and actively engage in postoperative lifestyle changes tend to achieve better results. Early identification of those at risk for suboptimal weight loss is critical. A proactive, individualized follow-up strategy—tailored to patient risk profile and potential timing of weight plateau or regain—can significantly enhance clinical outcomes. Personalized follow-up protocols are therefore key to maximizing long-term success [42,43].

## 5. Gastrointestinal Side-Effects

### 5.1. Dumping Syndrome

Dumping syndrome, common after RYGB, involves rapid gastric emptying into the small intestine, leading to early and late symptoms. Early-phase symptoms include nausea, diarrhea, dizziness, tachycardia, and abdominal discomfort associated with the ingestion of hyperosmolar foods, while late symptoms manifest as reactive hypoglycemia [44,45].

### 5.2. GERD and Dyspepsia

Post-SG GERD arises from increased intragastric pressure and reduced compliance of the stomach. ESG, by contrast, is less likely to induce GERD due to its preservation of gastric function [9,46]. Comparative studies highlight that approximately 30% of SG patients develop GERD postoperatively, whereas this complication is significantly less frequent after ESG [46]. GERD following SG is multifactorial in origin, involving anatomical and physiological changes induced by the surgery. The alteration of the gastric fundus during SG reduces the angle of His, thereby weakening the natural anti-reflux barrier and contributing to the increased incidence of postoperative GERD. Furthermore, the increase in intragastric pressure and decreased gastric compliance exacerbate reflux symptoms. In some cases, pre-existing hiatal hernias may worsen post-surgery, further contributing to GERD. Management strategies for GERD post-SG often include proton pump inhibitors (PPIs), lifestyle modifications, and, in severe cases, revisional surgery to address anatomical factors [47]. Recent studies suggest that careful patient selection and preoperative screening for reflux symptoms can reduce the risk of GERD following SG [48].

### 5.3. Nausea, Vomiting, and Pain

For IGB, the most common adverse events include balloon intolerance requiring early removal, persistent vomiting, gastroesophageal reflux, abdominal pain, gastric ulcers, and spontaneous deflation [49]. More severe complications such as gastric perforation, small bowel obstruction, impaction, and gastric dilation have also been reported [48]. In ESG, mild to moderate complications, such as dyspepsia, nausea, vomiting, and abdominal pain, are found in a significant percentage of patients and generally regress after 7 days [49]. These symptoms are often attributable to gastric inflammation and altered motility but are typically managed with pharmacological interventions [50,51].

The main serious adverse events of ESG are bleeding and abscess formation and are reported in only 2–3% of cases. This results in an excellent safety profile of the ESG procedure [49]. Pain following these procedures varies, with ESG associated with lower median pain scores compared to SG.

### 5.4. Nutritional Deficiencies

Nutritional deficiencies are frequent after bariatric surgery [11,12,13,14,15,16,17,18,19,20,21,22,23,24,25,26,27,28,29,30,31,32,33,34,35,36,37,38,39,40,41,42,43,44,45,46,47,48,49,50,51,52] and are influenced by the type of surgical procedure. These deficiencies are more frequent after malabsorption surgeries (such as BPD, RYGB and single anastomosis gastric bypass) than after restrictive procedures (such as LSG and LAGB) [53]. They are due to malabsorption of micronutrients like iron, calcium, and vitamin B12. ESG’s impact on nutritional status is comparatively minor, as it preserves GI continuity [3,11,54]. Other factors that determine a patient’s nutritional status are certainly pre-operative deficiencies [33,55] as well as the presence of GI symptoms such as vomiting or regurgitation, food intolerances, and incorrect eating habits [53,56,57].

Therefore, lifelong supplementation [52] and routine screening of vitamin and mineral status become necessary for patients undergoing surgery [58]. Supplementation should start upon discharge from hospital, usually 2–4 days after surgery [55]. Due to changes in absorption capacity, chewable or suckable pills are recommended for the first 3–6 months and then switching to oral supplements [11]. Minimal daily supplementation following bariatric surgery, depending on the procedure, should include 1–2 multivitamin and mineral supplements for adult, 1200–2400 mg of elemental calcium, ≥3000 IU of vitamin D (adjusted to therapeutic levels), and 250–350 μg of vitamin B-12 per day or 1000 μg of vitamin B-12 per week [11]. Endoscopic procedures, such as ESG, do not cause malabsorption. However, poor food choices, food intolerances and limited portions can lead to micronutrient deficiencies. Therefore, it is particularly important to detect and address these deficiencies early in order to prevent post-ESG deficiencies, so a multivitamin-mineral supplement to meet daily nutritional requirements is always recommended [52].

### 5.5. Long-Term Effects

The most significant long-term concern post-ESG is constipation, which has a more complex etiology, resulting from multiple factors [57]. The reduced gastric volume and altered gastrointestinal motility may result in a slowdown of the whole digestive process, leading to difficulty in bowel movements and even to harder stools. Additionally, changes in diet post-surgery, such as increased protein intake and reduced dietary fiber consumption, can exacerbate constipation. Some types of pain medications prescribed during the recovery phase can also contribute to decreased bowel motility. Dietary adjustments and probiotic supplementation have been suggested as effective strategies to mitigate these symptoms [59].

## 6. Gut Microbiota Alterations

Changes in the microbiota after bariatric surgery are currently proposed as one of many mechanisms explaining the positive clinical outcomes of BS [60,61]. Indeed, bariatric surgery may be considered a good model to study not only the pathophysiology of obesity and related diseases, but also the mechanisms involved in their improvement following weight loss [62].

To date, there are still few studies (as shown in Table 1) demonstrating that the composition of the gut microbiota is altered following RYGB, suggesting that weight reduction might influence the composition of the gut microbiota. However, weight loss may not be the only factor responsible for these changes. In fact, bariatric surgery not only improves hormonal and inflammatory status, but also induces numerous changes in the digestive tract that could explain the observed changes in microbiota ecology [63]. Recent studies have expanded the understanding of how bariatric surgery modulates the structure and metabolic function of the gut microbiota. Beyond early observations, several investigations have consistently reported increased microbial diversity, enrichment of Bacteroidetes and Proteobacteria, and reductions in Firmicutes after RYGB and SG, with these shifts correlating with improvements in glycemic control, inflammation, and body composition [53,64,65]. Long-term follow-up studies have confirmed persistent remodeling of microbial communities and fecal metabolites [66]. Systematic reviews further support the reproducibility of these postoperative microbiota changes across bariatric procedures, despite methodological heterogeneity among studies [62,67].

The works of Zhang et al. (2009) and Furet et al. (2010) [68,69], provided the first insights into microbiota changes after gastric bypass surgery. These studies were conducted on well-characterized cohorts of individuals with obesity, for whom detailed clinical data and dietary questionnaires were available, and whose findings were compared with those of lean control subjects. In the study by Zhang et al. [68] it was found that Firmicutes were dominant in normal-weight and obesity subjects, but were markedly reduced in post-RYGB patients. More importantly, functional differences in the microbiota were observed in subjects with obesity compared with those with normal weight or post-surgery. In fact, hydrogen (H2)-producing bacterial groups, such as Prevotellaceae, and H2-utilizing methanogenic Archea (both involved in energy extraction from indigestible polysaccharides) were present only in subjects with obesity and absent in lean subjects or after surgery. In addition, the levels of Gammaproteobacteria increased significantly after surgery. However, it is not possible to conclude whether these changes are the cause or the consequence of weight loss [68]. Moreover, Furet et al. [69] found that the *Bacteroides*/*Prevotella* ratio was lower in subjects with obesity than in control individuals and increased 3 months after surgery to remain stable thereafter. Importantly, this ratio was negatively correlated with corpulence traits (body weight, BMI, body fat mass, and serum leptin concentrations), but the correlation was strongly dependent on caloric intake. In contrast, in patients with obesity, increased levels of *Faecalibacterium prausnitzii* were directly associated with a reduction in low-grade inflammatory status independently from caloric intake; levels of *F. prausnitzii* were low in patients with obesity and type 2 diabetes at baseline and increased after surgery, highlighting that changes in the microbiota after surgery could depend on pre-surgical characteristics, particularly metabolic traits [68,69]. In both studies, it is not possible to determine whether the bacterial changes are due to changes in food intake and digestion, specific changes in the surgical procedure, or metabolic improvements. In fact, considering that bariatric procedures result in different rearrangements of the digestive tract, they probably have different effects on the modulation of the gut microbiota [63].

In the study by Medina D.A. et al., [70] the impact of dietary treatment, ESG and RYGB on the gut microbiota of subjects with obesity was compared. Compared with dietary treatment, changes in the gut microbiota were more pronounced in patients undergoing surgery, observing a proliferation of Proteobacteria. Interestingly, the abundance of Bacteroidetes was widely different six months after each surgical procedure. Furthermore, changes in weight, BMI and glucose metabolism positively correlated with changes in these two phyla in both surgical procedures [70]. Beyond bacterial composition, multiple studies highlight the role of microbial metabolites—such as SCFAs, secondary bile acids, betaine, and choline—in mediating metabolic improvements after BS [64]. In longitudinal analyses, both RYGB and SG resulted in marked changes in metabolite profiles, including reductions in acetate, butyrate, and propionate concentrations [65,66]. In addition to human clinical studies, further evidence comes from animal models. Preclinical studies in rodents undergoing bariatric procedures have similarly demonstrated surgery-induced alterations in gut microbiota composition, suggesting that these changes may occur independently of dietary factors and may contribute to the metabolic benefits of bariatric surgery [71,72,73]. More recently, Coimbra et al. conducted a systematic review, analyzing 18 clinical studies published in the last 10 years, with the aim of evaluating and characterizing the changes in the microbiota post BS [62]. It was observed that the predominance of intestinal bacterial phyla varied among the studies; however, most of them reported a higher proportion of Bacteroidetes, Proteobacteria, and greater diversity after BS. Results about the presence of Firmicutes, Bacteroidetes and the ratio of the two (F/B) were inconsistent, increasing or decreasing after RYGB and SG, compared with the one before surgery. In addition, a higher proportion of Actinobacteria was observed following RYGB. In contrast, the same was not identified when SG procedures were applied. Ultimately, the review showed that the abundance of genera and predominance of bacteria varied depending on the surgical procedure, with limited data regarding the impact on phyla [62]. Both SG and RYGB are associated with significant shifts in gut microbiota composition, including an increase in Bacteroidetes and a reduction in Firmicutes, which are linked to enhanced metabolic profiles [4,61].

Emerging evidences underscore the important role of the gut microbiota in shaping the physiological and metabolic outcomes of bariatric surgery. Notably, Zhang et al. [68] reported that Firmicutes, typically dominant in both lean and with obesity individuals, are significantly depleted following RYGB, indicating a profound shift in gut microbial composition after surgery. This observation is further supported by Voermans et al. [67], who showed that RYGB increases alpha diversity and promotes a rebalancing of the Firmicutes/Bacteroidetes ratio—although to a lesser extent than SG. Postoperative microbial profiles commonly include increases in *Escherichia coli* and *Bacteroides*, alongside expansion of the phylum Proteobacteria and the family Bacteroidaceae, coupled with a reduction in Firmicutes [68,71,72,73]. These changes are likely driven by both the anatomical modifications of the gastrointestinal tract and the resulting shifts in nutrient flow and substrate availability. Beyond intestinal effects, changes in the gut microbiota influence systemic pathways and neuro-hormonal mechanisms. As reviewed by Hamamah et al. [74], postoperative microbiota remodeling affects dopamine-regulated reward circuits and modulates the secretion of gut-derived hormones such as ghrelin, leptin, GLP-1, PYY, and CCK. Of particular interest is the consistent enrichment of *Akkermansia muciniphila*, a species associated with improved metabolic regulation. Of particular interest, the consistent postoperative enrichment of beneficial taxa such as *Akkermansia muciniphila* suggests a mechanistic connection between microbiota remodeling and improved metabolic regulation.

ESG, while less extensively studied, shows similar beneficial alterations in gut microbiota, supporting improved energy metabolism and inflammation reduction [49,61]. The minimally invasive nature of ESG helps maintain gut integrity, reducing dysbiosis compared to more invasive surgical interventions. Emerging evidence suggests that ESG facilitates microbiota changes conducive to better metabolic outcomes, such as reduced inflammation and enhanced SCFAs production. Taken together, these findings highlight the gut microbiota as a dynamic mediator of the effects of bariatric surgery and support its emerging role as a potential therapeutic target in the treatment of obesity and associated metabolic disorders. Figure 2 provides a visual summary of the progressive shift from a dysbiotic microbial profile typically observed in individuals with obesity to a more diverse and metabolically favorable composition following bariatric surgery or ESG. The diagram highlights the reduction in pro-inflammatory and energy-harvesting taxa (such as *Prevotellaceae*, methanogenic Archaea, and certain Firmicutes) and the concomitant enrichment of beneficial microbes including *Akkermansia muciniphila*, *Faecalibacterium prausnitzii*, and members of the Bacteroidetes phylum. These microbial transitions parallel the physiological improvements described in the literature—reduced inflammation, enhanced SCFA signaling, improved glycemic control, and healthier energy metabolism—illustrating how structural or endoscopic interventions reshape the gut ecosystem and contribute to metabolic recovery.

## 7. Nutritional Approaches

### 7.1. Before Bariatric Surgery/ESG Interventions

For the nutritional management of patients undergoing bariatric surgery, the American Society for Metabolic and Bariatric Surgery (ASMBS) and the international federation for the surgery of obesity and metabolic disorders (IFSO) provide specific guidelines regarding both the preoperative and postoperative periods [8].

It is essential to conduct a comprehensive assessment of the patient’s nutritional status, including the evaluation of potential nutrient deficiencies. Pre-operative weight loss is often recommended to reduce surgical risks and improve surgical accessibility even if it is not recommended to achieve baseline weight loss prior to ESG procedures. This can be achieved through a hypocaloric Mediterranean diet or a ketogenic diet (VLEKT). The latter determines similar results in terms of weight loss and change in body composition compared to the Mediterranean diet but in a shorter time (1 month versus 3 months) [75].

### 7.2. After Bariatric Surgery/ESG Interventions

After the bariatric surgery, patients gradually switch from a liquid diet to a pureed diet and finally to solid foods [8]. Kotzampassi and Lopez-Nava et al. also recommend keeping patients on a liquid diet for the first 2 weeks after the treatment [76,77]. This progression helps prevent complications such as dumping syndrome and facilitates the adaptation of the digestive system. Given the high risk of nutritional deficiencies due to reduced intake and absorption of nutrients, vitamin and mineral supplementation is crucial. This includes multivitamins, calcium, vitamin D, iron, and vitamin B12. It is important to ensure adequate protein intake to maintain muscle mass and promote healing. Generally, a daily intake of 60–80 gr of protein is recommended, adjusted based on individual needs [8].

Nutritional guidelines for ESG procedures are not yet available, so nutritional recommendations for bariatric surgery have been adapted. In the study conducted by Negi A. et al., a standardized dietary protocol was adopted for both IGB and other endoscopic gastroplasty procedures. Specifically, the protocol was developed for the first four weeks and detailed in terms of: meal type, caloric intake, examples of foods, and suggested meals [60].

Phase 1 (first week after procedure): A liquid-based diet with an estimated daily intake of approximately 400 kcal, favoring the consumption of fruit juices without added sugars, puréed vegetable soups, liquid yogurt, and sugar-free isotonic drinks. During this phase, it is crucial to consume small volumes at frequent intervals to avoid rapid gastric distention and minimize potential intolerance. Additionally, great attention should be given to hydration status, and the intake of water should be strongly encouraged.Phase 2 (second week): A diet ranging from 400 to 600 kcal per day is recommended. This phase begins when symptoms of gastric distress have improved, and the patient is better able to tolerate liquids. The volume of the diet is increased by introducing protein shakes, vegetable purees, and egg whites. Frequent meals in small volumes should be encouraged.Phase 3 (third week): The liquid diet should have facilitated postoperative healing and tissue repair, so the food’s consistency is gradually increased. Caloric intake reaches 800 kcal per day following a “soft” diet, which involves the gradual reintroduction of mechanically altered foods (e.g., ground meat, pureed, mashed) such as fruit compote, olive oil, and cooked egg whites.Phase 4 (fourth week): A transition toward the normal consistency of foods begins. Caloric intake is increased to 800–1200 kcal, and it is recommended to replace protein shakes with semi-solid or solid protein sources, such as chicken, fish, and fresh cheese.

Ultimately, it is recommended to follow a hypocaloric diet and to adopt strict healthy eating habits by implementing behavioral changes. Patients should be encouraged to keep track of their meals through a food diary and actively commit to consistently following a physical activity program. Constant monitoring with the healthcare team is essential to assess nutritional status and prevent long-term complications [8].

### 7.3. Long Term Maintenance

The classification of the bariatric procedures is considered from a nutritional perspective because the impacts of bariatric surgery on nutritional status are mostly related to the reduced stomach volume and decreased nutrient absorption [78].

Once the postoperative nutritional protocol is completed, patients should receive periodic counseling from a dietitian regarding long-term dietary recommendations to maximize the outcomes of the bariatric procedure and reduce the risk of late weight regain. The goal of dietary counselling should be to modify eating behaviors to adopt a healthy, nutrient-rich diet [78].

Adequate protein intake is considered protective against the loss of lean body mass during rapid weight loss, however protein consumption is often significantly reduced after BS (BS), particularly in the initial months following the procedure, primarily due to gastric intolerance to protein-rich foods. Current guidelines recommend a minimum protein intake of 60 g/day and up to 1.5 g/kg of ideal body weight per day following BSBS [79]. However, in some cases, higher protein intake (up to 2.1 g/kg) may be necessary. Given the challenges of achieving these targets with natural foods alone, the use of liquid protein supplements (30 g/day) is recommended as a means of supporting adequate protein intake in the first few months after BS. Current recommendations regarding protein intake and supplementation after BS are based on the results of a one-year prospective observational study showing an inverse relationship between protein intake and lean tissue loss, as a percentage of total weight loss, in 50 patients treated with RYGB (25 patients) or SG (25 patients) analyzed by dual-energy X-ray absorptiometry (DEXA) [80]. Negi et al. also proposed the bariatric plate: 50% protein, 20% carbohydrates (with low glycemic index) and 30% vitamins, minerals, and fibers, represented by fruits and vegetables [60]. To date, the available literature on long-term nutritional approaches post-ESG procedure is very limited. Few studies in the literature focus on the use of the Mediterranean diet or high-protein diets.

The high-protein diet is a frequently subscribed diet post-endoscopic therapy. The protein-based bariatric plate model by Cambi et al. describes the macronutrient and micronutrient composition of daily meals to promote and maintain weight loss over the long term [81].

However, the results of the study by Rueda-Galindo L. et al., showed that patients who followed a Mediterranean-style diet lost 14.2% more weight compared to those who followed a high-protein diet. Additionally, adhering to a Mediterranean-style diet was associated with greater long-term sustainability and a more gradual weight loss [82].

## 8. Conclusions

Bariatric surgery and endoscopic sleeve gastroplasty promote a profound shift from a dysbiotic to a more balanced and metabolically efficient gut microbiota, contributing to substantial improvements in weight, inflammation, and metabolic health. This transition mirrors the conceptual model illustrated in Figure 3, where surgical or endoscopic interventions, together with adherence to a Mediterranean dietary pattern, support the restoration of microbial diversity and metabolic homeostasis. Nutritional management remains essential throughout this process, ensuring adequate intake, preventing deficiencies, and enhancing long-term efficacy. Overall, integrating procedural approaches with targeted nutritional strategies and microbiota modulation offers a comprehensive and synergistic framework for the treatment of obesity.

## Figures and Tables

**Figure 1 nutrients-18-00108-f001:**
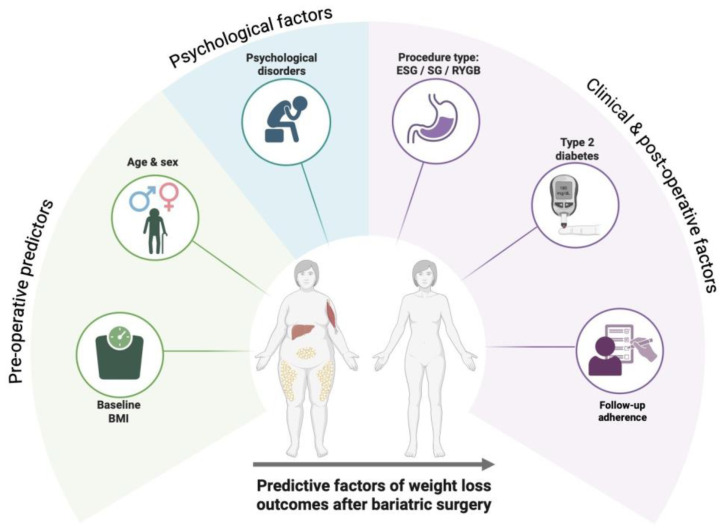
Main predictive factors influencing weight loss outcomes after bariatric surgery, including baseline BMI, age, presence of type 2 diabetes, psychological disorders, type of procedure, and adherence to postoperative follow-up.

**Figure 2 nutrients-18-00108-f002:**
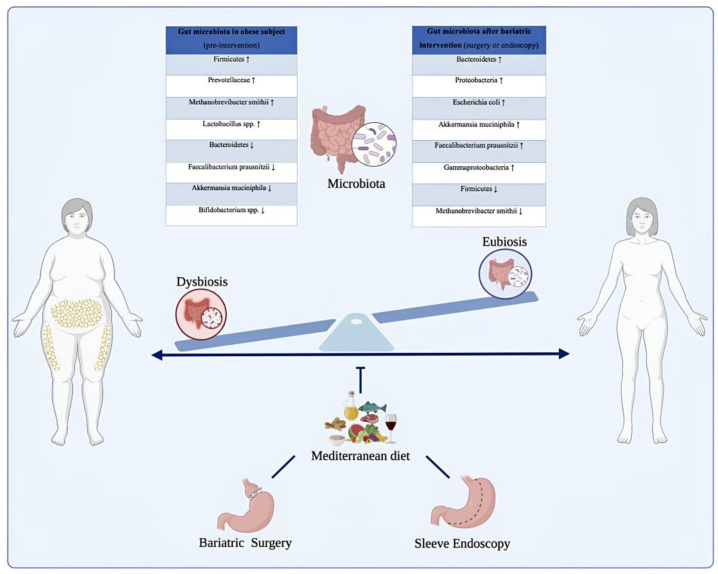
Modulation of the gut microbiota in individuals with obesity before and after bariatric or endoscopic interventions, supported by Mediterranean dietary patterns. The illustration depicts the transition from a dysbiotic intestinal microbiota, commonly observed in individuals with obesity, to a state of eubiosis following bariatric or endoscopic procedures, in conjunction with nutritional management. Before intervention, the gut microbiota is dominated by taxa associated with chronic low-grade inflammation and enhanced energy harvest, such as Firmicutes, Prevotellaceae, *Methanobrevibacter smithii*, and *Lactobacillus* spp., alongside a reduction in protective species including *Faecalibacterium prausnitzii*, *Bifidobacterium* spp., and *Akkermansia muciniphila*. Post-intervention, a profound microbial shift occurs, characterized by increased abundance of Bacteroidetes, Proteobacteria, *Escherichia coli*, *Faecalibacterium prausnitzii*, and *Akkermansia muciniphila*, resulting in improved metabolic and inflammatory profiles. The Mediterranean diet, rich in fiber, polyphenols, and monounsaturated fats, contributes to microbiota modulation and supports long-term weight loss maintenance and metabolic homeostasis. The up arrows indicate an increase while the down arrows indicate a decrease.

**Figure 3 nutrients-18-00108-f003:**
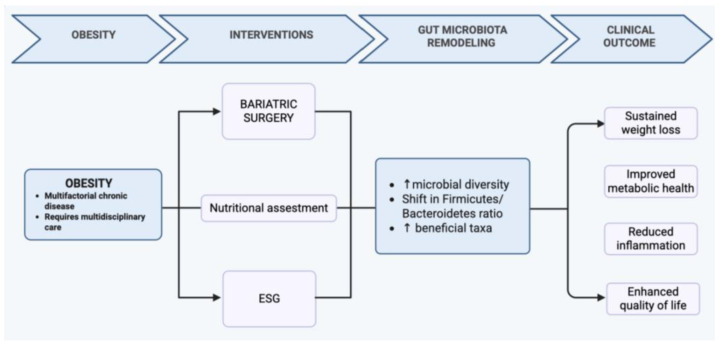
Conceptual model illustrating the synergistic effects of bariatric or endoscopic interventions and Mediterranean dietary adherence on gut microbiota restoration, metabolic homeostasis, and long-term obesity management. The up arrows indicate an increase of the values related to the condition.

**Table 1 nutrients-18-00108-t001:** Intestinal human microbiota changes after bariatric surgery.

Reference	Number of Patients	Type of Procedure	Time Span	Results
Zhang et al. [68]	n = 9 (3 normal-weight, 3 morbidly obese and 3 post gastric -bypass surgery)	Gastric—bypass surgery	18 months	Normal-weight: *Firmicutes*, **↑** Subjects with obesity: *Firmicutes*, *Prevotellaceae*, H_2_-utilizing methanogenic *Archaea* **↑** Post surgery: *Gammaproteobacteria*, **↑** *Firmicutes and Methanogens* **↓**
Furet et al. [69]	n = 43 (13 lean and 30 with obesity—7 of them also with T2D)	RYGB	6 months	(1) Bacteroides/Prevotella group was lower in subjects with obesity than in control subjects at baseline and increased after 3 months. (2) *Escherichia coli* increased at 3 months and inversely correlated with fat mass and leptin levels independently of changes in food intake; (3) Lactic acid bacteria and Bifidobacterium decreased at 3 months; and (4) *Faecalibacterium prausnitzii* was lower in subjects with diabetes and associated negatively with inflammatory markers at baseline and throughout the follow-up (3 and 6 months) after surgery independently of changes in food intake.
Medina et al. [70]	19 subjects	RYGB, SG or diet	12 months	In comparison to dietary treatment, changes in intestinal microbiota were more pronounced in patients underwent to surgery, with an increase in *Proteobacteria*. *Bacteroidetes* were different after six months of each surgical procedure.
Yu et al. [64]	20 subjects with obesity	Roux-en-Y gastric bypass (n = 10), sleeve gastrectomy (n = 10)	6 months	Bariatric surgery induced marked shifts in gut microbiota composition and fecal metabolomic profiles. Significant changes were observed in microbial diversity and in key metabolic pathways, including reductions in short-chain fatty acids and bile acid metabolites. These microbiota and metabolite alterations correlated with postoperative weight loss and improvements in metabolic parameters.
Shen et al. [65]	26 subjects with severe obesity	Gastric bypass or SG	12 months	Bariatric surgery tended to increase alpha diversity, and significantly altered beta diversity, microbiota composition, and function up to 6 months after surgery, but these changes tend to regress to presurgery levels by 12 months.
Juárez-Fernández et al. [66]	9 subjects	Vertical sleeve gastrectomy (VSG) (n = 6), biliopancreatic diversion (BPD) (n = 2) and gastric bypass (n = 1),	4 years	The phyla enriched in patients with obesity before surgery came predominantly from the phylum Cyanobacteria and the class Clostridia, whereas most of those in patients with obesity after surgery were enriched mainly from the class Choriobacteria. Bariatric surgery transformed the composition of the gut microbiota, showing a significant reduction in the phylum Firmicutes compared to the same patients before surgery, while the relative abundance of the phyla Proteobacteria and Lentisphaerae increased significantly in response to the bariatric procedure.

## Data Availability

Data is contained within the article.

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
