# Peer review of "Nutrients2026, 18(1), 108;https://doi.org/10.3390/nu18010108"

_nutrients, 2025, doi:10.3390/nu18010108_

Round 1

Reviewer 1 Report

Comments and Suggestions for Authors

This review is up-to-date and provides a comprehensive synthesis of all relevant topics in the field. It integrates the most recent evidence in a clear, structured, and analytically rigorous manner, ensuring that key aspects of symptoms, nutritional management, and gut microbiota after bariatric and endoscopic procedures are thoroughly addressed.

Author Response

We sincerely thank the Reviewer for this positive and encouraging comment. We are grateful for the recognition of the relevance, clarity, and analytical rigor of our review, as well as for the appreciation of its comprehensive and up-to-date synthesis of the literature. We are pleased that the Reviewer found our discussion of gastrointestinal symptoms, nutritional management, and gut microbiota changes after bariatric and endoscopic procedures to be thorough and well structured. This feedback strongly supports the value of our work and is greatly appreciated.

Reviewer 2 Report

Comments and Suggestions for Authors

Reviewer comments

The authors reviewed evidence on gastrointestinal side effects, gut microbiota changes, and nutritional management following bariatric surgery and endoscopic sleeve gastroplasty (ESG) in obesity treatment. They reported that both surgical procedures and ESG are associated with gastrointestinal symptoms and micronutrient deficiencies, while inducing significant shifts in gut microbiota that support metabolic improvement. The authors noted that ESG, although less invasive, produces microbiota changes comparable to bariatric surgery with a better safety profile. They emphasized that structured nutritional management, including supplementation and progressive diets, is essential to prevent deficiencies and maintain long-term outcomes. Overall, the authors highlighted the need for multidisciplinary, personalized approaches integrating procedures, nutrition, and microbiota modulation.

  1. Line 77-78 “Recent studies have shown that gut microorganisms contribute to the regulation of energy homeostasis and fat mass accumulation” The authors needs to add references related to this
  2. Line 82, explain the points with other published studies
  3. Line 321 bacteria name should be italic, please proofread wherever the authors used

Faecalibacterium prausnitzii

  1. Are there only 5 studies recently published, you can report more if you find it
  2. The term “narrative systematic review” is unclear because narrative and systematic reviews use different methods. Please clarify whether this work is mainly a narrative review with a structured search, or a full systematic review following PRISMA or similar guidelines.
  3. The description of the literature search is too short. Please add basic details such as the main keywords you used, the criteria for including or excluding studies, and how many papers were screened and finally included.
  4. The section repeats information about gut microbiota dysbiosis several times. Please combine these similar points so the text flows better and avoids unnecessary repetition.
  5. The BMI guidelines for bariatric surgery are mixed together for general obesity and metabolic disease. Please separate these two categories and explain them clearly so the text is easier to understand.
  6. The section describes ESG twice (lines 70–75) with similar points about it being minimally invasive. These can be combined for better conciseness and clarity
  7. The authors can made more attractive figures including other relevant text in the manuscript
  8. Almost all the references need to be modified based on the MDPI journal

Author Response

The authors reviewed evidence on gastrointestinal side effects, gut microbiota changes, and nutritional management following bariatric surgery and endoscopic sleeve gastroplasty (ESG) in obesity treatment. They reported that both surgical procedures and ESG are associated with gastrointestinal symptoms and micronutrient deficiencies, while inducing significant shifts in gut microbiota that support metabolic improvement. The authors noted that ESG, although less invasive, produces microbiota changes comparable to bariatric surgery with a better safety profile. They emphasized that structured nutritional management, including supplementation and progressive diets, is essential to prevent deficiencies and maintain long-term outcomes. Overall, the authors highlighted the need for multidisciplinary, personalized approaches integrating procedures, nutrition, and microbiota modulation.

Reply: We sincerely thank the reviewer for this positive and encouraging comment. We have made several changes to the text, hoping to address all the comments and suggestions received.

Line 77-78 “Recent studies have shown that gut microorganisms contribute to the regulation of energy homeostasis and fat mass accumulation” The authors needs to add references related to this

Reply: Thank you for the suggestion. We added recent and foundational references supporting the role of gut microbiota in energy balance and adiposity regulation (Lines 150-158, References 10-18).

Line 82, explain the points with other published studies

Reply: Thank you for the suggestion. The section has been expanded as suggested and the paragraph was also reorganized for clarity Lines 150-158, References 10-18).

Line 321 bacteria name should be italic, please proofread wherever the authors used

Reply: Thank you for the suggestion. We corrected all bacterial species and genera to the appropriate italicized format throughout the manuscript (e.g., Faecalibacterium prausnitzii, Akkermansia muciniphila, Escherichia coli).

Faecalibacterium prausnitzii. Are there only 5 studies recently published, you can report more if you find it

Reply: Thank you for the suggestion. As suggested, we expanded the text and the Table 1 by including additional recent studies that characterize gut microbiota changes following bariatric surgery. These new entries reflect contemporary findings from clinical and metagenomic research and provide a more comprehensive overview of postoperative microbial shifts (Lines 519-528).

The term “narrative systematic review” is unclear because narrative and systematic reviews use different methods. Please clarify whether this work is mainly a narrative review with a structured search, or a full systematic review following PRISMA or similar guidelines.

Reply: Thank you for this important clarification. We agree that the term “narrative systematic review” may be misleading, as narrative and systematic reviews follow different methodological frameworks. Our work is not a full systematic review conducted according to PRISMA guidelines. Instead, it is a narrative review supported by a structured literature search in PubMed and Scopus. We have modified the text also in the abstract (line 23).

The description of the literature search is too short. Please add basic details such as the main keywords you used, the criteria for including or excluding studies, and how many papers were screened and finally included.

Reply: Thank you for this useful comment. We expanded the Methods section to provide greater transparency regarding the literature search strategy (paragraph 2. Methods).

The section repeats information about gut microbiota dysbiosis several times. Please combine these similar points so the text flows better and avoids unnecessary repetition.

Reply: Thank you for this helpful observation. We revised the Introduction and Section 6 to remove redundancies regarding the description of gut microbiota dysbiosis in obesity. Overlapping sentences were combined into a single, more coherent explanation to improve flow and clarity.

The BMI guidelines for bariatric surgery are mixed together for general obesity and metabolic disease. Please separate these two categories and explain them clearly so the text is easier to understand.

Reply: We thank the reviewer for this remark. We revised the paragraph describing BMI-based eligibility criteria for bariatric surgery to clearly distinguish indications for individuals with obesity alone from those with metabolic disease. The updated text now follows the 2022 ASMBS/IFSO guidelines and improves clarity by separating the two categories and explaining their rationale (Lines 69-75).

The section describes ESG twice (lines 70–75) with similar points about it being minimally invasive. These can be combined for better conciseness and clarity

Reply: We Thank the reviewer for this helpful observation. We revised the introduction to remove redundant descriptions of endoscopic sleeve gastroplasty (ESG). The two overlapping sentences were merged into a single, concise description to improve clarity and readability (Lines 81-84).

The authors can made more attractive figures including other relevant text in the manuscript

Reply: Thank you for this suggestion. We have added Figure 2 and a diagram.

Almost all the references need to be modified based on the MDPI journal

Reply: Thank you for this suggestion. We modified the references according to MDPI journal.

Reviewer 3 Report

Comments and Suggestions for Authors

File attach

Comments on the Quality of English Language

Remove abbreviation repetition

Improve paragraph structure and merge single or 2-3 lines into proper paragraph

Author Response

Reviewer 3

We sincerely thank the reviewer for these constructive comments. We have made several changes to the text, hoping to address all the suggestions received.

Line 100, (BPD-DS), define it on first use, and remove repetition in lines 137-138

biliopancreatic diversion with duodenal switch (BPD-DS) to broaden therapeutic efficacy [15].

Reply: Thank you for this observation. We revised the paragraph to fully define biliopancreatic diversion with duodenal switch (BPD-DS) upon first mention and removed the repeated definition appearing later in the manuscript to improve clarity and avoid redundancy.

Line 123, 127, 133, 215, 300, 361 remove repetition

Roux-en-Y Gastric Bypass (RYGB) entails the formation of a small gastric pouch

Laparoscopic Sleeve Gastrectomy (LSG)

laparoscopic adjustable gastric banding (LAGB)

Reply: Thank you for this important observation. We revised the manuscript to ensure that all bariatric procedures are defined only upon first mention.

Lines 150-162. Write in a single paragraph

Reply: Thank you for the suggestion. The section corresponding to lines indicated has been revised and is now presented as a single cohesive paragraph to improve clarity and readability (Lines 280-290).

Line 163, and 214, remove repetition Endoscopic sleeve gastroplasty (ESG), Throughout the paper, define abbreviations on the first use, and later on, rely only on the abbreviation without brackets, instead of the definition with the bracketed abbreviation.

Reply: Thank you for this comment. We removed the repeated definition of endoscopic sleeve gastroplasty (ESG) at lines 163 and 214. ESG is now fully defined only at first use, and all subsequent occurrences rely solely on the abbreviation to improve clarity and avoid redundancy.

Line 180. Remove the blank space

Reply: Thank you for noticing this formatting issue. The extra blank space at line 180 has been removed to improve consistency and visual layout (Line 307)

Line 186, define EBMIL

Reply: Thank you for pointing this out. The acronym EBMIL has now been defined upon its first appearance as “Excess BMI Loss (EBMIL)” in the section on predictive factors (Line 312).

Line 239-240, check the highlighted, is it a typo error

The alteration of the gastric fundus during SG reduces the angle of His,

Reply: Thank you for noting this issue. The sentence referring to the alteration of the gastric fundus during SG has been revised for clarity (Lines 417-419).

Merge lines 248-255, 270-283, 285-293,

Reply: Thank you for the suggestion. We have merged the sentences as suggested in paragraphs 5.3, 5.4 and 5.5).

Lines 294-295, What BS means, if it is an abbreviation, then use in standard form throughout

the paper. Changes in the microbiota after bariatric surgery are currently proposed as one of many mechanisms explaining the positive clinical outcomes of BS.

Reply: Thank you for the comment. We clarified this by defining “bariatric surgery (BS)” at its first occurrence and using only the abbreviation consistently throughout the manuscript. Where appropriate, we also distinguished “bariatric and metabolic surgery (BMS)” according to current guidelines.

Lines 306-308, revise

They are studies conducted on precisely phenotyped 306 cohorts of patients with obesity (for

whom clinical data and diet questionnaires were available) that 307 compared their results with lean individuals as controls.
Reply: We thank the reviewer for highlighting this issue. The sentence has been rewritten to correct the typographical error and improve clarity and readability. The revised version now reads:

“These studies were conducted on well-characterized cohorts of individuals with obesity, for whom detailed clinical data and dietary questionnaires were available, and whose findings were compared with those of lean control subjects” (Lines 531-534).

Section 6. Gut Microbiota

Write all microbial names according to the standard format of species and genus on first use

and repetition

Merge individual sentences into proper paragraphs

Reply: Thank you for this important observation. All microbial names have been reviewed and corrected according to standard taxonomic formatting. Genus and species names (e.g., Faecalibacterium prausnitzii, Akkermansia muciniphila, Escherichia coli) are now italicized throughout the manuscript, while higher taxonomic categories (e.g., Firmicutes, Bacteroidetes, Prevotellaceae) are presented in regular font. We also corrected capitalization errors and ensured consistency across Section 6 and the entire manuscript.

Merge lines 403-412

Reply: Thank you for this important observation. We have merged the lines (Lines 831-843).

Lines 482-484, remove repetition

Bariatric procedures, particularly Roux-en-Y gastric bypass (RYGB) and sleeve gastrectomy (SG), remain the most effective interventions for severe obesity and related metabolic comorbidities. [Figure 1, 55,63].

Reply: Thank you for this important observation. We have modified the Conclusion section, removing repetitions.

Explain Figure 1 in the discussion, and in conclusion, state what is concluded from the study.

Reply: Thank you for this suggestion. Figure 1 has now been explicitly explained in the Conclusion section, where we describe how it illustrates the transition from dysbiosis to eubiosis and the interaction between surgical/endoscopic interventions and dietary patterns.

Remove references from the conclusion, write it in a single and precise paragraph instead of

multiple paragraphs containing statements from literature.

Reply: Thank you for this suggestion. The Conclusions section has been fully revised to provide a concise and integrated summary of the main findings of the review.

Better to add diagrams in the discussion to support the literature claims.

Reply: Thank you for this thoughtful suggestion. We have added an additional diagram (Figure 3), which visually summarizes the conceptual pathway linking obesity, bariatric surgery or ESG, nutritional assessment, microbiota remodeling, and clinical outcomes. This schematic representation complements Figure 1 by illustrating the sequence of interactions—from the multifactorial nature of obesity to intervention-driven microbial changes and their impact on metabolic health, inflammation, weight loss, and quality of life.

Round 2

Reviewer 3 Report

Comments and Suggestions for Authors

file attached

Comments on the Quality of English Language

The quality of English requires thorough revision to fix minor issues.